# Exploring Chemical Variability in the Essential Oils of the *Thymus* Genus

**DOI:** 10.3390/plants13101375

**Published:** 2024-05-15

**Authors:** Karim Etri, Zsuzsanna Pluhár

**Affiliations:** Department of Medicinal and Aromatic Plants, Institute of Horticultural Science, Hungarian University of Agriculture and Life Sciences, H-1118 Villányi Str. 29–43, 1118 Budapest, Hungary; pluhar.zsuzsanna@uni-mate.hu

**Keywords:** thymol, chemotype, essential oil, active compounds, chemodiversity

## Abstract

Thyme remains an indispensable herb today, finding its place in gastronomy, medicine, cosmetics, and gardens worldwide. It is highly valued in herbal remedies and pharmaceutical formulations for its antibacterial, antifungal, and antioxidant properties derived from the richness of its essential oil, which comprises various volatile components. However, climate change poses a significant challenge today, potentially affecting the quality of thyme, particularly the extracted essential oil, along with other factors such as biotic influences and the plant’s geographical distribution. Consequently, complex diversity in essential oil composition was observed, also influenced by genetic diversity within the same species, resulting in distinct chemotypes. Other factors contributing to this chemodiversity include the chosen agrotechnology and processing methods of thyme, the extraction of the essential oil, and storage conditions. In this review, we provide the latest findings on the factors contributing to the chemovariability of thyme essential oil.

## 1. Introduction

Thyme, belonging to the *Lamiaceae* family, comprises various species known for their medicinal and aromatic qualities [1]. It is indigenous to the Mediterranean region, particularly the Iberian Peninsula and Northwest Africa. The common thyme, or *Thymus vulgaris*, is extensively used in food flavoring, pharmaceuticals, and personal care products, contributing significantly to its economic importance [2]. Meanwhile, local species like *Thymus pannonicus* (Pannonian thyme) or *Thymus capitatus* (Conehead/Mediterranean thyme; Spanish oregano) may not have the same level of commercial cultivation but still possess medicinal properties, and they may contain unique bioactive compounds or have specific therapeutic properties that make them valuable for medicinal purposes [3,4].

Despite its significance, the classification of *Thymus* is complex, presenting challenges in accurately identifying species due to variability and hybridization [5]. Considering this complexity, da Silva et al. [6] evaluated the genetic relationships among *Thymus* taxa using AFLP markers, which supported the accepted taxonomy based on morphological traits and essential oil content. Furthermore, *Thymus* taxa display variations in essential oil yield and composition. In the study of György et al. [7], *T. vulgaris* clones represented five different chemotypes and were distinguished using 12 ISSR primers, where thymol chemotype showed the most distinct separation. The study showed the potential of using molecular markers in the breeding and selection of *T. vulgaris*, being able to differentiate different chemotypes. Different major compounds are identified in *Thymus holosericeus* populations from various Ionian Islands in Greece [8]. Similarly, five *Thymus* species from Section Hyphodromi were found to have distinct chemotypes based on their essential oil composition, revealing variations in the abundance of sesquiterpenoids, aromatic compounds, and major components such as thymol and carvacrol [9]. These differences in essential composition between *Thymus* taxa lead to the distinction of various chemotypes, with the understanding that such variations may be attributed to genetic diversity and ecological factors [10].

In the following sections, we focus on the chemovariability of thyme essential oil by presenting factors that encompass genetic and ontogenetic effects, geographical variations, climatic influences, agrotechnological factors, as well as the influence of various processing methods.

## 2. Literature Research Methodology

Bibliographic research was carried out using reputable research database engines such as Google Scholar, Scopus, PubMed, SpringerLink, and Web of Science. Most of the consulted journals are highly indexed and peer-reviewed, including *Essential Oil-Bearing Plants*, *Agronomy*, *Horticulturae*, and *Industrial Crops and Products*. E-books were consulted as well, primarily “Thyme—The Genus *Thymus*”, in addition to referring to international standards organizations such as ISO and Ph. Eur. To obtain more accurate literature results, the following keywords were primarily used: “*Thymus* genus”, “Thyme”, “Chemodiversity”, “Chemotype”, “Factors”, “Essential oil”, and “Terpenes”.

The entire procedure followed for the literature research is presented in Figure 1. After inputting the keywords, several literature pieces were obtained from the databases. Approximately half of them were removed due to duplication reasons, articles focusing on other medicinal plants in their experimental part, or them being old publications. We also aimed to use the most recent results. A screening was conducted later, and irrelevant studies were excluded, followed by the retrieval of some literature, mainly the oldest ones, according to their date of publication, due to insufficient recent results. Finally, the remaining literature was assessed for eligibility. Studies were excluded if they focused on secondary metabolites not found in the essential oil of thyme, if they were superficial studies about thyme essential oil without focusing on its volatile components, or if they were articles with insufficient data, language problems, etc. In conclusion, a total of 147 pieces of literature were used in the current review out of the 723 initially identified from the databases.

## 3. Taxonomy and Essential Oil Production of Thyme

### 3.1. Taxonomy of Thyme

The *Thymus* genus, consisting of around 250 taxa, includes 214 species and 36 subspecies [11]. The *Thymus* genus is subdivided into eight sections based on geographical distribution and morphological characteristics.

The Micantes section encompasses the North African woody species *T. riatarum*, *T. satureioides*, and *T. caespititius*. Morphologically, they are erect plants with flat, globous, long oblong-obovate leaves and spiciform inflorescence [12,13]. α-Terpineol is considered a major active component in *T. caespititius* according to Miguel et al. [14], Pinto et al. [15], and Neves et al. [16]. High variability in the essential oil composition of *T. riatarum* in Morocco has been observed: Fadli et al. [17] revealed the dominance of Borneol as a major component, while Boubaker et al. [18] indicate the dominance of γ-terpinene and p-cymene.

Endemic to the Iberian Peninsula, the Mastichina section includes *T. mastichina* subsp. *mastichina*, *T. mastichina* subsp. *donyanae*, which are characterized generally by a high amount of 1,8-cineole, while the *T. albicans* species show linalool as a major compound in its essential oil in addition to 1,8-cineole. Morphologically, they show capituliform inflorescence with holotrichous stems and flat, lanceolate-to-obovate leaves [13,19,20,21].

The *Piperella* section represents a unique species: *T. piperella*, endemic to Valencia and its surrounding areas in Spain. It has similar leaves and stem morphology to the Mastichina section species, but it has a verticillaster inflorescence. p-cymene and carvacrol are considered the major active constituents in this species [13,22,23].

The *Teucrioides* section, originating from the Balkan Peninsula, is characterized by revolute, ovate, or triangular-ovate leaves and a verticillaster inflorescence. It comprises three different species: *T. teucrioides*, *T. hartvigi*, and *T. leucospermus* [13]. The three species show p-cymene as one of the main constituents within a population collected from Greece and Albania, as reported by Pitarokili et al. [24].

The Pseudothymbra section occurs in the Iberian Peninsula and North Africa. Morphologically, they are erect plants with linear revolute leaves and a capituliform inflorescence. It includes nine species: *T. lotocephalus*, *T. villosus*, *T. longiflorus*, *T. membranaceus*, *T. moroderi*, *T. munbyanus*, *T. bleicherianus*, *T. funkii*, and *T. antoninae.* All these species show a distinguishable number of active compounds. *T. munbyanus* subsp. *ciliates* from Algeria show carvacrol as a main constituent in its essential oil [13,25,26].

The *Thymus* section occurs in the Western Mediterranean region, and the plants are erect or radicant with revolute leaves and spiciform or globose inflorescences. It includes the most important species in thyme: *T. vulgaris*, *T. zygis*, and *T. willdenowii* [13]. *T. vulgaris*, commonly known as garden thyme, is a bushy, evergreen subshrub with small, highly aromatic gray-green leaves and clusters of purple or pink flowers. Several chemotypes are observed in this species, but generally thymol is considered the main constituent in the essential oil [27,28,29,30].

The Hyphodromi section consists of plants that are subtended and rooting with revolute or flat leaves and a capituliform inflorescence. The section encompasses the entire Mediterranean region and includes 60 species [13]. *T. algeriensis* is one of the popular species in this section with high chemical polymorphism; in Tunisia, it shows 1,8-cineole and camphor as major compounds [31], while in Algeria, in the same location but at different altitudes, *T. algeriensis* indicates a different chemical profile: thymol type and terpinyl acetate type [32].

The Serpyllum section includes around 120 species and occurs within the genus area. Morphologically, the plants are woody, with leaves ciliated at the base and a spiciform inflorescence. The well-known wild thyme or *T. serpyllum* belongs to this section, exhibiting creeping stems with pink to purple flowers. Concerning the chemical profile, thymol is considered the main active compound in its essential oil [13,33,34].

### 3.2. Essential oil Biosynthesis in Thyme

Thyme plants have been found to produce volatile oil in thyme glandular hairs, primarily carried out by peltate glandular trichomes (PGTs), which are responsible for the biosynthesis and accumulation of monoterpenes [35].

In *Lamiaceae*, glandular trichomes contain specialized secretory cells where volatile compounds are synthesized and stored within a subcuticular space, facilitating their retention in a liquid state until release [36]. The principal constituents of essential oils (EOs) are synthesized through three biosynthetic pathways: terpenes are generated via the plastidic methylerythritol phosphate pathway (MEP) and the cytosolic mevalonic acid pathway (MVA), while phenylpropenes are synthesized through the shikimic acid pathway [37,38].

The isoprenoid units (C5), dimethylallyl pyrophosphate (DMAPP), and isopentenyl pyrophosphate (IPP) are provided, respectively, via the MEP and MVA pathways. In plastids, they combine to form geranyl pyrophosphate (GPP, C10), the primary precursor for monoterpenes and diterpenes. Conversely, in the cytosol, they produce farnesyl pyrophosphate (FPP, C15), the precursor for sesquiterpenes [39,40]. Terpene synthases (TPSs) catalyze the formation of these terpenes’ basic skeletons, constituting a diverse class of enzymes responsible for the vast structural variety of mono- and sesquiterpenes [40,41]. Examples of compounds generated by TPSs include thymol, carvacrol, geraniol, linalool, camphor, and β-caryophyllene. [42]. Figure 2 illustrates the molecular structure of the most significant terpenes found within the *Thymus* genus.

### 3.3. Chemical and Biological Properties of Essential Oil Active Compounds

Thymol and carvacrol are commonly regarded as the primary bioactive constituents in thyme essential oil. However, thyme oil contains a complex mixture of other compounds that contribute to its overall aroma, flavor, and therapeutic effects.

#### 3.3.1. Chemical and Physical Properties of Thymol

Thymol is a colorless crystalline compound classified as a monoterpene phenol. Chemically, it is identified as 2-isopropyl-5-methylphenol with the formula C_10_H_14_O [43]. Thymol is produced through the aromatization of γ-terpinene to p-cymene, which undergoes hydroxylation to produce thymol [44]. Thymol is characterized by low solubility in water (1 g/liter at 20 °C) but has high solubility in alcohols and organic solvents [45,46]. It has a melting point around 50 °C [47] and a density of 0.96 g/cm^3^ at 25 °C [48]. Thymol has a strong flavor, but, at a suitable concentration, it can have a pleasant flavor with antibacterial efficiency (0.2 mg/mL of thymol) as described by Kerekes et al. [49].

#### 3.3.2. Chemical and Physical Properties of Carvacrol

Carvacrol, or 5-isopropyl-2-methylphenol, has the same chemical formula as thymol, making them structural isomers. This means that the hydroxyl group is at the ortho position of the benzene ring in carvacrol, whereas it is in the meta position for thymol [50]. It has a density of 0.991 ± 0.11 g/cm^3^ and a surface tension of 60.38 ± 0.08 mN/m at 30 °C [51]. Carvacrol, like thymol, has low water solubility [52].

Carvacrol is more soluble in water from 25 to 35 °C compared to thymol, while at higher temperatures, between 40 and 45 °C, thymol becomes more soluble [53].

#### 3.3.3. Biological Activities of Thymol and Carvacrol

All the active compounds present in the essential oil contribute to the medicinal properties of thyme, encompassing antioxidant, anti-inflammatory, anticarcinogenic, antibacterial, antifungal, and antiparasitic effects [54,55].

*T. vulgaris* (35.42% thymol and 28.79% p-cymene) has been found to have synergistic effects with tetracycline in reducing colorectal cancer cell viability, along with high antibacterial potential against *Staphylococcus aureus* (Gram-positive) and *Klebsiella pneumoniae* (Gram-negative) [56]. It also exhibits efficacy against various foodborne pathogens like *Bacillus cereus* (Gram-positive) and *Salmonella enterica* subsp. *Enterica serovar* (Gram-negative), showing its effectiveness in food preservation in the study by Sateriale et al. [57], which used *T. vulgaris* containing 44.43% thymol and 39.39% limonene.

*T. vulgaris* (48.96% thymol and 29.15% p-cymene) has recently been found to have antioxidant properties by protecting blood telomeres and anti-inflammatory effects by reducing pro-inflammatory cytokines in chronologically aged mice [58]. Furthermore, it reduced paw edema by 89.59% within 3 h after the application of *Thymus leptobotrys* Murb essential oil (73.68% Carvacrol) in the study by Oubihi et al. [59]. Another investigation confirmed the antioxidant and anti-inflammatory properties of *T. vulgaris* essential oil by reducing liver damage induced by renal ischemia-reperfusion in rats [60].

Arancibia et al. [61] demonstrated significant antiparasitic effects of thyme essential oil on plant-parasitic nematodes, achieving a 90% reduction in nematode numbers after 8 h of exposure in vitro. *T. serpyllum* (59.93% Carvacrol) essential oil demonstrated effective antiparasitic effects on Demodex mites, with carvacrol identified as a key active ingredient contributing to its efficacy [62].

#### 3.3.4. Chemical Synthesis Methods of Active Compounds: Focus on Thymol

The natural sources of active compounds remain important; however, today, the use of synthetic routes for their production has developed extensively, presenting a valuable avenue for developing novel compounds with improved properties for various industrial and medicinal purposes [63].

Thymol, a natural phenolic monoterpene, has been chemically synthesized through various methods. In 1920, Max Phillips and H. D. Gibbs [64] described a synthesis starting from p-cymene, involving nitration, reduction, sulfonation, and subsequent treatments to produce thymol. In 1932, an alternative approach was introduced by Niederl and Natelson [65], which involved the catalyzed condensation of propylene with m-cresol, facilitated by sulfuric acid; following this, the acid group was eliminated, resulting in the formation of thymol. Nowadays, industrial-scale production commonly uses the Bayer process, which involves the liquid-phase alkylation of m-cresol with propene at a high temperature and pressure using “active aluminum” catalysts, yielding thymol [66,67]. Indeed, subsequent studies have investigated diverse methodologies for m-cresol alkylation, employing a range of catalysts and reaction conditions. For instance, these include the utilization of propylene alongside aluminum 3-methylphenoxide, as reported by Grabowska et al. [68], as well as the application of microwave-assisted reactions in conjunction with carbonized sulfonic acidic resin catalysts, as highlighted by Ali and Gaikar [69]. In 2023, Mesbah et al. [70] efficiently synthesized the thymol from m-cresol through alkylation using isopropanol and ZSM-5 acid catalysts.

### 3.4. Authenticity of Thyme Essential Oil

#### 3.4.1. Methods for Detecting Active Compounds in Essential Oils

The authenticity of thyme essential oil can be accurately assessed using various analytical techniques. One of the most popular is gas chromatography-mass spectrometry (GC-MS), which enables the identification and quantification of a wide range of compounds present in complex mixtures like plant extracts, providing valuable information on their chemical composition [71,72]. This technique allows for the analysis of bioactive compounds, alkaloids, terpenoids, flavonoids, glycosides, and other secondary metabolites in herbal plants, aiding in the discovery of potential therapeutic agents [71,73].

Other methods have been employed to detect active compounds in thyme such as high-performance liquid chromatography (HPLC), which separates and quantifies compounds based on their interaction with a stationary phase and a mobile phase [74,75]. Spectroscopic techniques, including infrared (IR) and nuclear magnetic resonance (NMR) spectroscopy, are also used to analyze the chemical structure of compounds in thyme. They provide information about functional groups and molecular structure [76].

#### 3.4.2. Regulations and Standards for Thyme Essential Oil

The methods discussed above are the best way to confirm the authenticity, detect any adulteration, or identify contamination in thyme essential oil. These can be ensured through regulations and standards governing the production and marketing of essential oils from the main industrial thyme species.

##### The International Organization for Standardization (ISO)

The ISO, recognized for its global relevance in standards, specifies the physical requirements of essential oil extracted from *Thymus vulgaris* and *Thymus zygis* (thymol type). It is required to have a mobile liquid appearance, with a color ranging from yellow to red, and possess an aromatic and phenolic odor with a spicy base. The required chemical profile is illustrated in Table 1, which shows the minimum and maximum percentages of active compounds in thyme. Specifically, the thymol content is specified to be a minimum of 35% and a maximum of 55% (ISO 19817:2017) [77].

##### The European Pharmacopeia (Ph. Eur)

The Ph. Eur (11.0), recognized as a legal reference for standards and quality control of pharmaceutical products, also establishes standards for thyme products. *Thymi herba*, which refers to the leaves and flower parts separated from the dried stems of *Thymus vulgaris* L. and *Thymus zygis* L., are required to contain a minimum of 12 mL/kg of essential oil (anhydrous drug), of which 40% should comprise the sum of thymol and carvacrol contents (0865 (07/2014)) [78]. For *Serphylli herba* (*Thymus serphyllum* L.), known as wild thyme, a minimum of 3.0 mL/kg of essential oil (dried drug) is set as the standard requirement (1891 (07/2014)) [78].

The Ph. Eur also sets standards for the content of active compounds for *Thymi typo thymolo aetheroleum*, which describes the essential oil obtained from the fresh flowering aerial parts of *Thymus vulgaris* L. and *Thymus zygis*. The required chemical profile for Ph. Eur. 11.0 is presented in Table 1, where minimum standards slightly differ from those set by ISO 19817:2017 [78]. Specifically, the minimum thymol content is determined at 35% for ISO (19817:2017), whereas it is 37% for Ph. Eur (11.0).

## 4. Genetic and Ontogenetic Factors

### 4.1. Chemodiversity between Thymus Taxa

The essential oil composition within the *Thymus* genus displays significant variability. In the study conducted by Trendafilova et al. [79], *Thymus moesiacus*, *T. jankae*, *T. vandasii*, *T. longicaulis*, and *T. sibthorpii* share linalool as the primary component, but show variations in the concentrations of other prominent components such as geranyl acetate, geraniol, and linalyl acetate. Aksit et al. [80] identified thymol, p-cymene, eucalyptol, camphor, α-pinene, and carvacrol as the main components in the essential oils of *Thymus pectinatus*, *T. convolutus*, and *T. vulgaris*. Furthermore, *Thymus atticus*, *T. leucotrichus*, and *T. striatus* are characterized by high amounts of sesquiterpenoids, primarily β-caryophyllene, while *T. thracicus* exhibited a mixed mono-/sesquiterpene chemotype with β-myrcene, cis-sabinene hydrate, τ-cadinol, and elemol as its major components [9,79]. *Thymus kotschyanus* var. *kotschyanus* and var. *glabrescens* reveal distinct chemotypes, characterized by major compounds such as thymol, carvacrol, p-cymene, and γ-terpinene [81]. Similar findings regarding these components were reported by Hanoglu et al. [82] in the case of *Thymus capitatus.*

Moreover, different chemotypes were identified in *Thymus pannonicus*, each characterized by specific dominant compounds. Some chemotypes showed main monoterpenes like thymol, p-cymene, and γ-terpinene, while others contained compounds such as linalool, geraniol, and linalyl acetate. Sesquiterpenes, including germacrene-D, β-caryophyllene, and β-bisabolol, are also present in certain chemotypes [83]. Lemon thyme, scientifically identified as *Thymus citriodorus*, exhibits distinct chemical constituents, as evaluated by Jurevičiūtė et al. [84]. Noteworthy compounds include geraniol (36.6%), geranial (17.5%), nerol (11.5%), and neral (10.2%).

### 4.2. Intraspecific Variability

Intraspecific variability exerts a considerable influence on the composition of thyme essential oil. A study on 103 Korean populations of *Thymus quinquecostatus* revealed different chemotypes characterized by dominant compounds in the essential oil, including thymol, geraniol, geranyl acetate, and linalool. The chemical polymorphism observed among thyme populations was associated with the genetic diversity of individuals [85]. Substantial variability was observed in the ‘Standard Winter’ cultivar of *Thymus vulgaris*, showing diverse chemical compositions of the essential oil and in phenolic acids (caffeic acid, rosmarinic acid and p-Coumaric acid) and flavonoids contents [86].

Llorens-Molina et al. [87] investigated the chemodiversity of 85 populations of *Thymus vulgaris* in Spain, ultimately revealing five distinct profiles based on the essential oil composition as shown presented in Figure 3: camphane skeleton, oxygenated sesquiterpenic fraction, 1,8-cineole-rich with camphor and borneol, camphor and terpinen-4-ol, and linalool chemotype. A similar study was conducted on 21 populations of *Thymus zygis* L., leading to the distinction of seven main chemotypes [88]. High chemical variability in essential oil content was observed among seven populations of *Thymus caramanicus* in Iran, classifying the populations into three chemotypes (carvacrol, carvacrol/thymol, thymol/carvacrol) [89]. This variation can be an indicator of the importance of genetic diversity in the chemodiversity of thyme. In fact, the variability in essential oil composition can pose challenges in standardizing raw materials, but it also provides opportunities for breeding and the development of valuable components.

### 4.3. Impact of Phenological Stage

The composition of essential oil in thyme undergoes significant changes based on the plant’s growth stage, with higher yields and compositions observed during the blooming stage compared to the pre-flowering stage [90,91,92]. In *Thymus armeniacus*, Carvacrol exhibits a significantly higher amount at 100% of the flowering phase compared to 50%, with linalool, borneol, thymol, and β-caryophyllene showing a similar trend [90]. During the flowering stage, essential oils in *Thymus numidicus* are rich in oxygenated monoterpenes (82.11% compared to 67.2% in the pre-flowering stage), such as thymol and p-cimene-7-ol, known for their antioxidant and antimicrobial properties [93].

The chemical composition of essential oil also varies depending on the year of harvesting and the phenological stage of the plants; thymol and carvacrol were the main compounds in the essential oil of *T. vulgaris* studied by Moisa et al. [94], but their amounts varied depending on the year of harvesting and the flowering stage. Antimicrobial activity, correlated to the amount of compounds detected in the essential oil, was found to be higher during the seed set stage in particular for *Thymus fedtschenkoi* Ronniger [91].

Several other studies have investigated the variability of compounds along the flowering stage. In *Thymus × citriodorus*, the highest percentage of terpinolene was observed during the flowering stage [95]. In *Thymus piperella*, the concentration of carvacrol peaks at the beginning of the flowering stage, a trend similarly observed in γ-terpinene but with a slightly lower rate, while the yield of p-cymene decreases during the flowering stage [96]. In Kermanian thyme (*Thymus caramanicus* Jalas.), thymol was the main compound of the oil in all phenological stages, with the highest essence and thymol yields obtained during the fruit set stage [97].

In summary, thyme essential oil composition varies significantly across growth stages, with the flowering stage being particularly influential. Key compounds like carvacrol, terpinolene, and thymol show significant fluctuations as discussed. These findings underline the importance of considering phenological stages during the harvesting to optimize thyme oil quality for diverse applications.

## 5. Environmental Factors

### 5.1. Geographical Variation of the Essential Oil Composition

The essential oil composition of thyme varies depending on geographical location; this variability has been presented in several papers. Toncer et al. [95] conducted a study on *T. citriodorus* grown in the semi-arid continental climate of southeastern Anatolia, Türkiye, revealing variations in essential oil composition during different phenological stages. Major compounds identified included terpinolene, α-terpineol, linalool, bornyl acetate, and borneol. In contrast, *T. citriodorus* cultivated in Iran was dominated by geraniol, geranial, and neral [98].

Further insights into the geographical influence on thyme essential oils were provided by Chbel et al. [99], who analyzed *Thymus vulgaris* essential oils from Morocco and France. Variances in major constituents, such as borneol, α-terpinol, and carvacrol (31.04%, 15.16% and 7.13%, respectively) in Moroccan oils compared to thymol, ortho-cymene, and γ-terpinene (35.77%, 17.23% and 8.05%, respectively) in French oils, were observed. Additionally, the anti-microbial and free radical scavenging activities were found to be higher in *T. vulgaris* cultivated in France. Satyal et al. [100] analyzed different chemotypes of *Thymus vulgaris* essential oils and identified 20 distinct chemotypes based on their compositions. Specifically, the essential oil from Nyons, France, exhibited the linalool chemotype, whereas the oil sample from Jablanicki, Serbia, displayed the geraniol chemotype. Additionally, the essential oil from the Pomoravje District, Serbia, was characterized by the sabinene hydrate chemotype, while that from Richerenches, France, evidenced the thymol chemotype. Lastly, the types of essential oils in thyme differ not just between countries but also among nearby regions. This suggests that environmental factors have a substantial impact on these differences.

### 5.2. Climate and Environmental Influences on the Essential Oil Composition

The essential oil composition of thyme is influenced by various abiotic factors, contributing to its diverse chemical profile. Abiotic factors such as moisture levels, salinity, temperature, sunlight exposure, heavy metal content, and soil composition all play a role in shaping the composition of thyme essential oils.

#### 5.2.1. The Effect of Drought Stress

Drought stress has been found to impact the essential oil composition of thyme. Studies indicate that drought stress can result in significant increases in the concentration of 1,8-cineole in wild *Thymus vulgaris* L. in the eastern Iberian Peninsula [101]. Similarly, an increase in 1,8-cineole was observed in *T. daenesis* and *T. vulgaris* cultivated under drought stress conditions [102]. Moreover, it was revealed that the percentages of γ-terpinene and p-cymene in the oil were higher, while the percentage of thymol was significantly reduced under drought stress [102,103].

#### 5.2.2. The Effect of Salinity Stress

Salinity has been found to positively impact the essential oil composition of thyme. It influences phenolic compounds, leading to an increase in total phenolic monoterpenes such as γ-terpinene and p-cymene, leaf flavonoid content, and various phenolic acids. Indeed, concentrations of cinnamic acid, gallic acid, and rosmarinic acid were significantly elevated under salinity stress, while thymol was significantly reduced [104,105]. It has also been demonstrated that at a NaCl level of 100 mM, thymol and carvacrol increased in *Thymus vulgaris* cv varico 3 [106]. However, in a study on *Thymus maroccanus*, increased salinity did not affect the essential oil content [107].

#### 5.2.3. The Effect of Temperature and Humidity

A study on *Thymus transcaucasicus* revealed that elevated air temperature and lower humidity positively influenced herbal productivity, leading to increased essential oil, carotenoids, and polyphenols. The essential oil composition showed variations based on temperature and humidity conditions: at 25 °C, only borneol and α-terpineol increased, whereas at 15 °C, 1,8-cineole, thymol, carvacrol, geraniol, and geranyl acetate showed an increase. Furthermore, borneol and carvacrol reached their peaks at 50% humidity, whereas linalool and geraniol increased at 90% humidity [108]. Interestingly, thymol biosynthesis, controlled by gene expression (DXR and TPS1), exhibited a similar reducing trend as thymol under cold stress treatment [109].

#### 5.2.4. The Effect of Light

Sunlight intensity also plays a significant role in thyme. Higher light intensities result in increased oil yields and greater leaf thickness, contributing to enhanced biomass production in *Thymus vulgaris*. Interestingly, the concentration of thymol in the essential oil remains unaffected by variations in light incidence [110,111]. In a field study examining the impact of solar ultraviolet (UV) radiation, on three Thymus species (*T. daenensis*, *T. fedtschenkoi*, *T. vulgaris*), enriched UV led to varied effects on phenolic compounds in the essential oil, with some, like thymol, carvacrol, and γ-terpinene, increasing, and others decreasing [2].

#### 5.2.5. The Effect of Heavy Metals

The effect of heavy metals on the essential oil composition of thyme has also been studied. One study investigated the impact of excessive concentrations of heavy metals such as nickel, copper, and zinc on *Thymus vulgaris* plants. It was observed that the antioxidant capacity and total phenolic compounds decreased with an increased concentration of these heavy metals [112]. In fact, high concentrations of Zn, Pb, and Cr have been observed, affecting the quality of the produced thyme [113,114].

#### 5.2.6. The Effect of Soil Properties

The essential oil composition of thyme is influenced by soil properties such as humus, pH, organic content, nitrogen, phosphorus, and the presence of certain elements. The presence of humus in the soil positively influences the accumulation of essential oil in *Thymus pulegioides*. Additionally, soil composition with elevated levels of sulfur and mobile phosphorus, along with lower sodium content, promotes the prevalence of the carvacrol chemotype in *T. pulegioides* [115]. In *Thymus vulgaris*, the application of humic acid to the soil enhances the essential oil content, with the highest percentage of thymol observed under higher levels of humic acid [116]. The thymol content in *T. kotschyanus* is also influenced by the sodium absorption ratio (SAR), organic content, and the percentage of nitrogen and calcium carbonate in the soil, as demonstrated through multivariable regression and RDA methods employed by Aminzadeh et al. [117]. *Thymus pannonicus* and *T. praecox* exhibit different essential oil chemotypes, with *T. pannonicus* thriving in soil characterized by slightly acidic or neutral pH and elevated levels of certain elements such as K, Ca, Mg, Mn, Fe, Co, and Cr [118].

## 6. Agrotechnical Factors

In addition to the abiotic factors, biotic influences, such as microbial activity and interactions with other plant species, also contribute to the chemical variability in the essential oil of thyme.

### 6.1. Effect of Biofertilizers

Biotic factors, including the application of bio-fertilizers and stress-modulating nanoparticles, significantly influence the composition of thyme essential oil. Amani Machiani et al. [119] demonstrated that the use of arbuscular mycorrhizal fungi (AMF) and chitosan nanoparticles (CHT) as bio-fertilizers, either individually or in combination, enhances both the quantity and quality of thyme essential oil, particularly thymol, p-cymene, and γ-terpinene. Furthermore, the application of Thiobacillus bio-fertilizer and treatment with superabsorbent were found to impact the content of thymol, caryophyllene, and borneol in *Thymus vulgaris* and *T. daenesis* [120]. Abdel-Hamid et al. [121] revealed that the inoculation of a bacterial consortium, acting as a plant growth-promoting bacteria, could affect the oil composition in *T. vulgaris*, resulting in a higher thymol content compared to the control.

### 6.2. Effect of Plant Density

In addition, thyme plant density within a specific area is another factor influencing the essential oil composition of thyme. MalekMaleki et al. [122] found that reducing plant density in *Thymbra spicata* L. led to a decrease in total phenol and flavonoids. Optimal row spacing between 20 and 30 cm has been suggested for achieving the best morphological and biochemical plant parameters [122,123]. Conversely, Punetha et al. [124] demonstrated that *Thymus linearis* of the Himalayan thyme prefers narrower spacing and higher plant densities, resulting in increased yields of essential oil and improved quality.

The composition of thyme essential oil is highly influenced by a variety of factors, as discussed previously and summarized in Table 2, showing the effect of different abiotic and biotic factors on essential oil composition of thyme species. Under the different mentioned factors, *Thymus vulgaris* shows an increased rate of thymol in the essential oil, in case of increased salinity at 150 mM NaCl [125] and at 100 mM NaCl [126] using substrate that contain 100 gm^−2^ of humic acid, using biofertilizers such as AMF [119] or PGPB [121], and in the case of increasing shade rate by using shade nets with a shade index of 40% [127]. However, thymol rate decreased in case of drought stress [102,103]. In fact, in response to stress, plants may exhibit a resistance mechanism by partially closing their stomata, which restricts the absorption of a reduced amount of CO_2_ by the plant, directing it primarily towards the synthesis of essential secondary metabolites crucial for preventing cellular oxidation [128].

A high variability has been noticed in the concentration of the other active compounds in the different factors mentioned in Table 2, which can illustrate how these factors collectively contribute to the complexity and variability of thyme oil. While, in general, biotic factors such as AMF or PGPB and the application of humic acid showed higher active compound rates, this can be an effective way to obtain a higher quality of essential oil in thyme.

## 7. Impact of Processing Methods

### 7.1. Drying Methods

The drying methods employed for thyme significantly influence the composition of its essential oil. In the case of *Thymus vulgaris*, drying at 35 °C was identified as the most effective for essential oil content compared to natural drying, drying at 40 °C, or the freeze-dried method. However, it is noteworthy that the thymol content reached its peak at 40 °C [130]. Furthermore, a combination of convective pre-drying at 40 °C and vacuum-microwave finish drying at 240 W resulted in higher volatile compounds when compared to convective drying, vacuum-microwave drying, and freeze-drying methods [131].

The freezing stabilization method (to −18 °C in 4 h) has been shown to provide better preservation of the components of the essential oil of *T. vulgaris* compared to the freeze-drying method (between −50 and 20 °C in 24 h) or air drying (between 38 and 45 °C) as demonstrated by Usai et al. [132]. Lyophilization of flowering stems in *T. vulgaris* has been found to result in better preservation of thymol and carvacrol compared to natural drying and oven drying at 30 °C. However, lyophilization may result in lower levels of p-cymene compared to natural drying and oven drying at 30 °C [133].

### 7.2. Extraction Methods

Different methods of essential oil extraction have been shown to significantly influence the composition of thyme essential oil. A comparison of distillation methods, specifically, water-steam distillation and hydrodistillation, as detailed by Wesolowska et al. [134], demonstrated significant differences in the chemical composition of garden thyme oil, resulting in variations in the dominant components such as thymol, carvacrol, p-cymene, and γ-terpinene. Khokhlov et al. [135] revealed that using the steam distillation method for producing essential oil in both thymol and linalool chemotypes of *Thymus vulgaris* resulted in a decrease in their main compounds (linalool and thymol, respectively) compared to the hydrodistillation method using Clevenger apparatus. This alteration also affected the composition of other components. Conversely, Sadjia et al. [136] found that steam distillation of *Thymus pallescens* yielded slightly higher contents of monoterpenes, sesquiterpenes, and oxygenated sesquiterpenes compared to steam diffusion and hydrodistillation methods.

Supercritical fluid extraction (SFE) is an efficient and selective alternative to conventional extraction methods that uses supercritical fluids, such as CO_2_, as the extracting solvent [137]. Oszagyan et al. [138] revealed that by using SFE, they obtained a higher amount of carvacrol but a lower amount of thymol compared to steam distillation. After comparison with different extraction conditions, Kutta et al. [139] proposed the optimal parameter profile for SFE-CO_2_ to obtain a rich extract with volatile compounds for *T. vulgaris* (17 MPa, 45 °C, 30 min) and for *T. pannonicus* (24 MPa, 55 °C, 20 min). In contrast, a higher amount of monoterpenes in *T. vulgaris* was detected using simultaneous distillation extraction (SDE) compared to SFE [140]. Additionally, hydrodistilled essential oil of *T. vulgaris* showed a higher thymol ratio (48–50%) than the SFE samples (10–15%) [138]. Another modern method for extracting organic compounds is solid-phase microextraction (SPME), which is also a solvent-free method. It uses stationary phases on a fiber to extract analytes for GC analysis, classified as direct or headspace methods [141]. Sárosi and Ruff [142] advised optimizing the SPME method according to each plant species and revealed higher content of active compounds, particularly thymol, in the hydrodistilled oil than when using the SPME method.

Interestingly, contrasting studies have demonstrated that the extraction method, whether classical hydro-distillation, microwave hydro-distillation, or supercritical fluid extraction, did not influence the composition of thyme essential oil, as indicated by András et al. [143]. Similarly, Gedikoğlu et al. [144] observed a similar composition of thyme essential oil using both extraction methods: hydrodistillation and microwave-assisted extraction.

### 7.3. Storage Conditions

Storage temperature has been shown to affect the composition of essential oils extracted from thyme. Storing the oil at room temperature (25 °C) initially resulted in a reduction in the compounds. However, after 3 months under the same storage conditions, the levels of thymol and carvacrol increased. In contrast, storing the essential oil in a refrigerator preserved its primary quality with minimal alterations to the composition [145]. According to Mohammadian et al. [146], thymol content in *Thymus vulgaris* remained stable during the first 3 months of storage; however, it subsequently decreased under different conditions studied, including room temperature, refrigerator (4 °C), and freezer (−20 °C).

Thymol, γ-terpinene, and carvacrol have been better preserved after freezing and storing at −20 °C compared to the drying and freeze-drying methods, which were stored at 20 °C and in the dark [132]. Similar results regarding the increase in thymol and carvacrol have been reported by Rowshan et al. [147] under all the studied storage conditions: ambient temperature (25 °C), refrigerator (4 °C), and freezer (−20 °C).

## 8. Conclusions

In summary, the essential oil composition of thyme is a complex interplay of various factors, geographical and environmental influences, plant development traits and processing methods. The inherent variability within the *Thymus* genus, coupled with specific chemotypes and their responses to environmental stimuli, emphasizes the rich diversity observed in thyme essential oils. This variability presents both a challenge and an opportunity for industries dependent on thyme oil, necessitating careful consideration of factors such as growth stages during harvesting and appropriate processing methods. As a result, understanding the complexity of the interaction of these elements is essential for optimizing thyme oil quality and ensuring its suitability for diverse applications.

## Figures and Tables

**Figure 1 plants-13-01375-f001:**
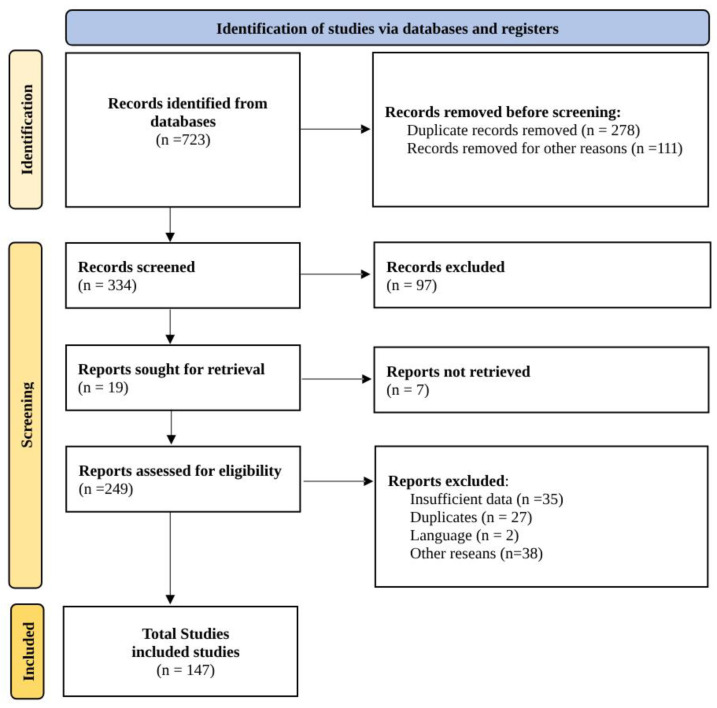
PRISMA diagram followed for the literature search.

**Figure 2 plants-13-01375-f002:**
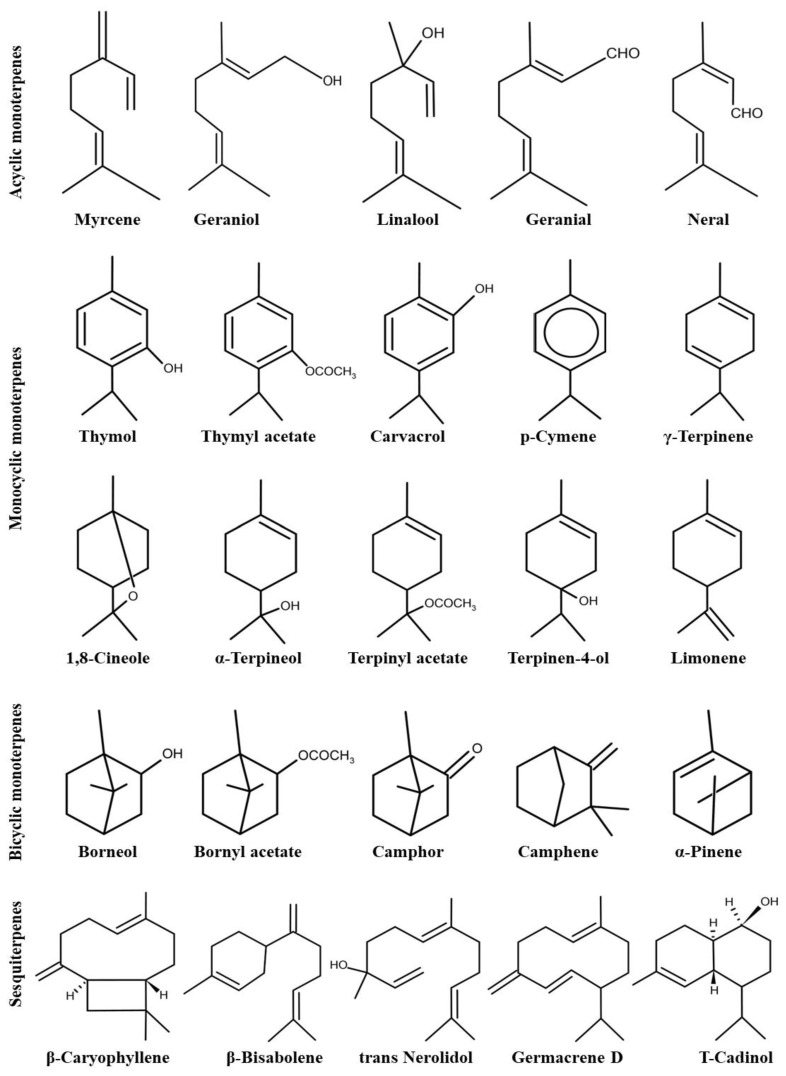
Chemical structures of the most important terpenes found among the genus of *Thymus* (After Stahl-Biskup [42]).

**Figure 3 plants-13-01375-f003:**
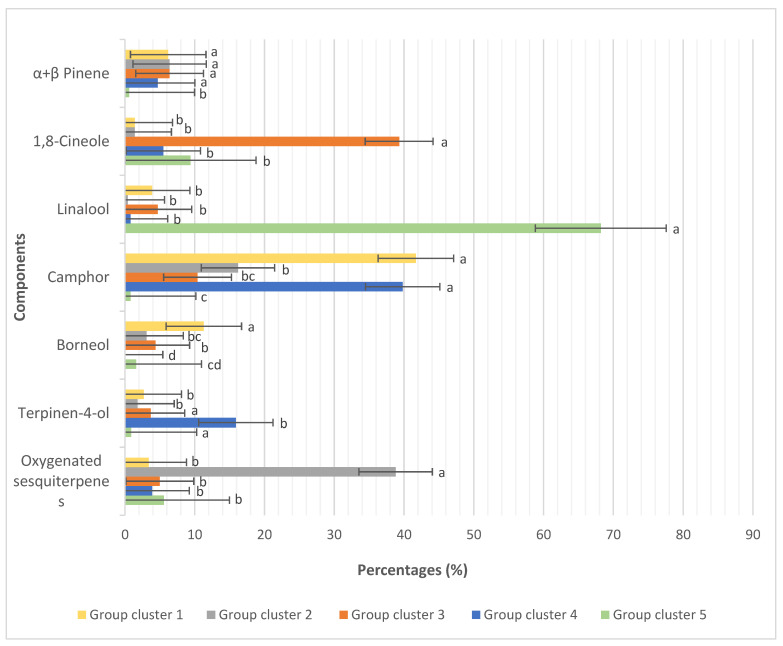
Average percentage (%) of the main components in the essential oil of *Thymus vulgaris* among the defined groups from the Cluster Analysis. Different letters within the same component indicate a significant difference between the group clusters, as determined by the Tukey’s HSD test (*p* < 0.05) (After Llorens-Molina et al., 2020) [87].

**Table 1 plants-13-01375-t001:** Chromatographic profile standards of the essential oil of *Thymus vulgaris* and *Thymus zygis* (thymol type) as mentioned in ISO 19817:2017 [77] and in Ph. Eur. 11.0, 10,000 (01/2023) [78].

	ISO 19817:2017		Ph. Eur. 11.0, 10,000 (01/2023)	
Component	Minimum (%)	Maximum (%)	Minimum (%)	Maximum (%)
α-Thujene	0.50	1.50	0.20	1.50
α-Pinene	0.50	2.50	ns	ns
β-Myrcene	1.00	2.80	1.00	3.00
α-Terpinene	0.90	2.60	0.90	2.60
γ-Terpinene	4.00	13.00	4.00	12.00
p-Cymene	14.00	28.00	14.00	28.00
Linalool	0.50	6.50	1.50	6.50
Terpinen-4-ol	0.10	2.50	0.10	2.50
Thymol	35.00	55.00	37.00	55.00
Carvacrol	0.50	5.50	0.50	5.50
β-Caryophyllene	0.50	4.00	ns	ns
trans-Sabinene hydrate	<0.01	0.50	ns	ns
Carvacrol methyl ether	0.10	1.50	0.05	1.50

ns: not specified.

**Table 2 plants-13-01375-t002:** Effect of abiotic and biotic factors on essential oil composition (%) of thyme species.

Factor	Reference	*Thyme* spp.	Treatment	α-Thujene (%)	α-Pinene (%)	Camphene (%)	p-cymene (%)	1,8-Cineole (%)	α-Terpinene (%)	γ-Terpinene (%)	Linalool (%)	Borneol (%)	α-Terpineol (%)	Thymol (%)	Carvacrol (%)	β-Caryophyllene (%)
Drought	[103]	*T. vulgaris*	Calus control	1.70	2.20	3.00	7.70	1.40	1.50	14.20	1.20	1.80	1.00	13.20	13.50	7.20
Callus Polyethylene Glycol (PEG)	2.00	1.60	5.10	10.80	1.60	1.10	19.30	1.10	2.10	1.30	9.80	9.60	3.10
[102]	*T. vulgaris T. daenensis*	Normal irrigation	1.08	1.32		9.92	1.34	1.95	11.33		2.00		51.99	3.80	2.37
Slight drought irrigation	1.10	1.17		11.79	1.19	2.05	12.66		2.35		41.46	4.77	2.71
Salt	[125]	*T. vulgaris*	Control	12.09	6.67	12.67		0.01	0.06	2.89	12.37	0.16		15.09	3.13	2.77
150 mM NaCl	4.15	6.06	17.35		0.05	0.08	4.75	16.45	0.38		20.06	4.48	4.10
[126]	*T. vulgaris*	Control	1.70	2.20	6.20	15.10			1.10	22.10	5.70		29.40	3.40	2.40
100 mM NaCl	2.04	1.52	3.80	8.70			0.30	24.20	10.90		32.70	1.10	1.70
Light	[127]	*T. vulgaris*	Unshade (Open field condition)	0.18	0.12	0.07	2.80	0.05	0.45	3.49	0.32	0.23	0.06	8.05	0.65	0.27
40% shade	0.28	0.17	0.07	3.60	0.03	0.55	4.04	0.21	0.19	0.03	9.35	0.73	0.33
Temperature	[108]	*T. transcaucasicus*	15 °C		1.82		1.14	7.10	0.88	0.06	6.34	11.70	17.37	11.18	18.05	0.02
25 °C		1.99		0.91	3.34	0.47	0.03	3.59	25.31	33.77	8.54	5.46	0.17
Humidity	[108]	*T. transcaucasicus*	50%		1.98		0.57	1.04	0.08	1.81	6.59	42.05	14.57	2.60	14.47	0.40
90%		2.98		1.04	1.10	0.27	3.33	7.64	30.65	15.82	3.12	12.31	0.16
Humic acid	[116]	*T. vulgaris*	Control	0.23	0.25	0.23	3.46	0.58	0.35	2.24	1.73	3.12	0.17	73.08	6.18	
100 gm^−2^	0.26	0.27	0.25	4.24	0.53	0.48	2.40	1.58	3.42	0.12	74.15	6.20	
Arbuscular Mycorrhizal Fungi (AMF)	[129]	*T. satureioides*	Control		2.06	0.65			8.47		1.67	5.93		16.55	18.50	2.26
AMF+		1.27	0.46			8.37		1.62	16.73		16.75	6.05	2.57
[119]	*T. vulgaris*	Control	1.66	1.21	0.88	16.35	3.02	1.22	13.11	2.06	0.80	0.07	35.64	2.94	3.23
AMF+	1.34	0.56	0.65	16.72	3.15	1.18	12.83	2.09	0.70	0.05	38.74	2.99	3.55
Plant Growth-Promoting Bacteria (PGPB)	[121]	*T. vulgaris*	Control	2.93	0.84				8.38	23.29			4.95	29.85		
PGPB+	3.49	1.00				7.42	22.20			1.64	33.31

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
