# Peer review of "Exploring Chemical Variability in the Essential Oils of the Thymus Genus"

_plants, 2024, doi:10.3390/plants13101375_

Round 1

Reviewer 1 Report

Comments and Suggestions for Authors
    • Could you please provide a section discussing the authenticity of Essential Oils derived from the Thymus genus.
    • Figure 1: requires redrawing for better clarity and understanding.
    • Please include the full names of AMF and PGPB as footnotes below Table 1.
    • Graphic abstract is highly recommended. 
Comments on the Quality of English Language
  • There seems to be a typographical error that needs correction.

Author Response

Please see the attachment ''Reviewer1''

Reviewer 2 Report

Comments and Suggestions for Authors

In general, I found the manuscript well structured and interesting. It has a significant potential to become a highly cited article.

However, some minor revisions must be done.

Firstly, Figure 1 must be replaced with a better quality figure. I saw it was taken from another article and you have provided a reference. However, it does not look good. You should draw an original figure, using a program for chemical structures or something else.

Secondly, check the punctuation of Table 1 title. In my view the percent symbol should be after the numbers in the table. Explain better in the text the meaning of "control", "unshaded", which are met in Table 1.

Finally, you should explain the search procedure. It is a good option to include a PRISMA figure. Only 75 ref. are included. You must explain why, etc. In my view more ref. must be included to support the data.

Author Response

Please see the attachment ''Reviewer 2"

Reviewer 3 Report

Comments and Suggestions for Authors

The manuscript by Karim et al. reviewed the chemovariability of thyme essential oil, and factors including encompass genetic and ontogenetic effects, geographical variations, climatic influences, agrotechnological factors, and processing methods were discussed. In my opinion, the manuscript can be accepted after minor revisions. Please see my comments to improve the manuscript.

The scope of this journal determines that this review will focus on the effects of biological and environmental factors on the Thymus Genus and its essential oils. However, in the actual application of essential oils, in addition to providing unique flavors to products, their biological effects such as antibacterial and antioxidant properties depend more on key active substances. As the author mentioned in the article, thymol, γ-terpinene, p -cymene, and carvacrol contribute to the medicinal properties of thyme. However, these key substances already have corresponding chemical synthesis methods, which weakens the significance of direct extraction from plants. I recommend the authors introduce the chemical properties of key compounds in thyme, such as chemical synthesis methods.

In section 2, there is a lack of introduction to the plant biosynthesis of thyme. Meanwhile, there is a lack of in-depth discussion of different thyme taxa. The author only discussed the conclusions of previous studies and needs more summary.

The article also lacks a comprehensive discussion of processing methods. Section 4.3-4.5 only introduced the most basic extraction and storage technologies, lacking a more comprehensive discussion. Some important extraction and stabilization technologies need to be discussed, such as new extraction technologies (such as supercritical extraction), encapsulation technologies, etc., which are worthy of introduction.

Figure 1 needs to be redrawn to have a better vision.

Table 1, AMF, PGPB should use full names.

Author Response

Please see the attachment ''Reviewer 3"

Round 2

Reviewer 1 Report

Comments and Suggestions for Authors

No issues detected